# Diastereo- and atroposelective synthesis of N-arylpyrroles enabled by light-induced phosphoric acid catalysis

Lei Dai [1], Xueting Zhou [1,2], Jiami Guo[1,2], Xuan Dai[1], Qingqin Huang[1,2] & Yixin Lu [1,2] ✉

The C−N axially chiral N-arylpyrrole motifs are privileged scaffolds in numerous biologically active molecules and natural products, as well as in chiral ligands/catalysts. Asymmetric synthesis of N-arylpyrroles, however, is still challenging, and the simultaneous creation of contiguous C−N axial and central chirality remains unknown. Herein, a diastereo- and atroposelective synthesis of N-arylpyrroles enabled by light-induced phosphoric acid catalysis has been developed. The key transformation is a one-pot, three-component oxo-diarylation reaction, which simultaneously creates a C−N axial chirality and a central quaternary stereogenic center. A broad range of unactivated alkynes were readily employed as a reaction partner in this transformation, and the N-arylpyrrole products are obtained in good yields, with excellent enantioselectivities and very good diastereoselectivities. Notably, the N-aryl-pyrrole skeletons represent interesting structural motifs that could be used as chiral ligands and catalysts in asymmetric catalysis.

Axially chiral structural motifs are commonly present in bioactive compounds, and they are also widely used as chiral ligands or catalysts in asymmetric catalysis[1–8]. Therefore, atroposelective synthesis of axially chiral molecules has become one of the most-investigated research areas in recent years[9–16]. As part of our ongoing research efforts, we recently became interested in atroposelectively constructing an axially chiral axis between a five-membered heterocycle and an aryl ring[17], since these molecules are valuable, yet their asymmetric synthesis is inherently challenging and has not been well studied[18,19]. In this context, N-arylpyrrole skeletons are often found in natural products, chiral ligands, and catalysts[20–25] (Fig. 1a), therefore, catalytic asymmetric synthetic methods to access this type of molecules would be of great significance and highly appealing. There are only a handful of reports describing catalytic asymmetric synthesis of N-arylpyrroles up to date. Utilizing catalytic asymmetric Paal−Knorr reaction, Tan and co-workers achieved highly atroposelective synthesis of arylpyrroles[26]. Through remote control, the same group constructed axially chiral N-arylpyrroles via a desymmetrization or kinetic resolution strategy[27]. Through a chiral-at-metal rhodium Lewis acid-

catalyzed atroposelective electrophilic aromatic substitution, Houk, Meggers, and co-workers achieved atroposelective synthesis of axially chiral N-arylpyrroles[28]. Very recently, Szpilman et al. reported a copper- and chiral nitroxide-catalyzed kinetic resolution of axially chiral N-arylpyrroles[29]. Given the importance of N-arylpyrrole compounds, and the scarcity of methods for their atroposelective synthesis, we decided to devise an efficient asymmetric synthetic approach to access these molecules.

In devising a catalytic atroposelective synthetic method to prepare N-arylpyrroles, we opted to make use of unactivated alkyne substrates, as alkynes are a family of pivotal and sustainable feedstocks for pharmaceutical and agrochemical industries[30–39]. When the asymmetric construction of axially chiral molecules is concerned, there are numerous examples that alkynyl substrates were synthetically manipulated for the creation of atropoisomers[40]. In 2004, Shibata and co-workers reported the first asymmetric synthesis of axially chiral compounds via an iridium-catalyzed [2 + 2 + 2] cycloaddition[41]. Subsequently, the construction of axial chirality from alkynes via transition metal catalysis has been extensively investigated[42–53]. On the other

[1]Department of Chemistry, National University of Singapore, 3 Science Drive 3, Singapore 117543, Singapore. [2]Joint School of National University of Singapore and Tianjin University, International Campus of Tianjin University, Binhai New City, Fuzhou 350207 Fujian, China. ✉e-mail: chmlyx@nus.edu.sg

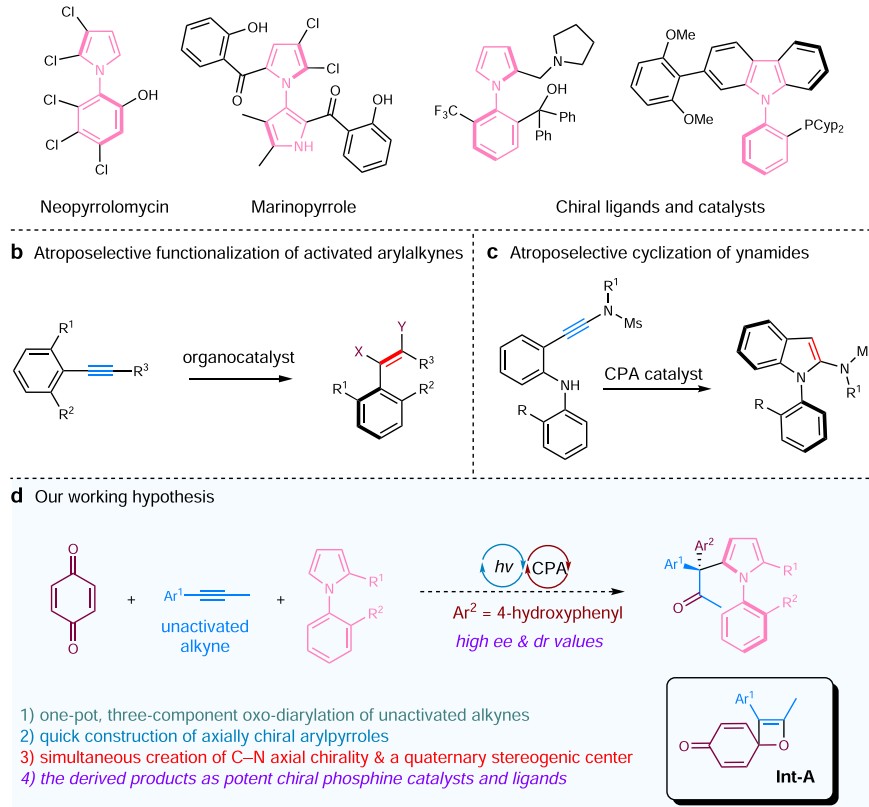

**Fig. 1 | Background and our working hypothesis. a** Representative examples containing axially chiral *N*-arylpyrrole skeleton. **b** Atroposelective functionalization of activated arylalkynes. **c** Atroposelective cyclization of ynamides. **d** Our working hypothesis. Cyp cyclopentyl, Ms methanesulfonyl, CPA chiral phosphoric acid, dr diastereomeric ratio, ee enantiomeric excess.

hand, the examples on organocatalytic atroposelective functionalization of alkynes are much less. In general, the alkynyl moieties in the substrates are subjected to two types of transformations; through a catalytic addition reaction to form axially chiral styrenes[54–59] (Fig. 1b), or undergoing an annulation reaction to yield biaryl atropisomers[60–67] (Fig. 1c). Apparently, the employment of alkynes as one of reaction partners for the construction of axial chirality would be more desirable, as simple and unactivated alkyne substrates are readily available, while the alkynyl substrates are not, and their preparation often requires extra synthetic steps. In our projected reaction of utilizing alkyne substrates for the creation of axial chirality, we envisioned that light-induced Paternò-Büchi [2 + 2] reaction[68–71] between an alkyne and a quinone would generate the crucial spiro-oxetene intermediate (**Int-A**). Under phosphoric acid catalysis and in the presence of pyrrole substrates, the ring-opening of oxetane to form *p*-quinone methide (*p*-QM) intermediate[72–77] and the subsequent nucleophilic addition with *N*-arylpyrroles are anticipated to deliver axially chiral *N*-arylpyrrole products (Fig. 1d). Herein, we report an asymmetric preparation of axially chiral *N*-arylpyrroles, via an atroposelective oxo-diarylation of unactivated alkynes enabled by light-induced phosphoric acid catalysis.

## Results

### Optimization of the reaction conditions

We initiated our investigation by running a three-component reaction involving alkyne **1a**, benzoquinone **2a** and *N*-arylpyrrole **3a** in the presence of different chiral phosphoric acid (CPA) catalysts under 440 nm Kessil LEDs irradiation (Table 1). With the employment of CPAs **5a–5e**, the reaction proceeded smoothly, however, the enantioselectivities

were poor (entries 1–5). When CPA **5f** with a bulky triphenylsily group was used, the reaction virtually did not take place (entry 6). We were delighted to discover that the utilization of CPAs **5g** & **5h** led to dramatic improvement on the enantioselectivity of the reaction (entries 7 and 8). A solvent screening was then followed. Among different solvents examined, only dichloromethane and *n*-butyronitrile were comparable to acetonitrile (entries 9–12). We next lowered the reaction temperature to further enhance stereoselectivities of the reaction. When the reaction was performed in acetonitrile at −42 °C, 89% ee and 19:1 dr were obtained (entry 13). Since the melting point of acetonitrile is at −45 °C, we then used a mixture of acetonitrile and *n*-butyronitrile (melting point −112 °C) to run the reaction at lower temperatures. After some experimentations, we established the optimal reaction conditions; when the reaction was performed in a mixed solvent system (acetonitrile/*n*-butyronitrile = 5:1) at −50 °C, the desired product was obtained in 85% yield, with 20:1 dr and 92% ee (entry 15).

### Substrate scope

The generality of the reaction was subsequently investigated (Fig. 2). The suitability of different alkynes was evaluated first (Fig. 2a). Alkynes bearing alkyl chains with the length ranging from one (methyl) to five (*n*-pentyl) were well tolerated, and regiospecific products with excellent diastereo- and enantioselectivities were obtained in good yields (**4a–4e**). The alkyl moiety in the alkyne substrates possessing a benzyl or a phenylethyl group were also found to be suitable (**4f−4g**). Interestingly, a free hydroxyl group in the alkyne was also found applicable (**4h**). Moreover, naphthyl alkynes also turned out to be good substrates (**4i−4j**). Both terminal and diaryl alkynes turned out to be suitable substrates (**4k** & **4l**). However, no product was formed with the

## Table 1 | Optimization of the reaction conditions[a]

Reaction scheme: 1a + 2a + 3a → 4a with CPA 5 (5 mol%), 440 nm Kessil LEDs, 0 °C to -50 °C. Ar$^1$ = β-naphthyl; Ar$^2$ = 4-hydroxyphenyl.

5a, R = Ph
5b, R = 3,5-(CF$_3$)$_2$C$_6$H$_3$
5c, R = 9-anthracenyl
5d, R = 9-phenanthrenyl
5e, R = 1-pyrenyl
5f, R = SiPh$_3$
5g, R = 2,4,6-(i-Pr)$_3$C$_6$H$_3$
5h, R = 2,6,-(i-Pr)$_2$,4-Ad-C$_6$H$_2$

| Entry | CPA | Solvent | Yield[b] | dr[c] | ee[d] |
|---|---|---|---|---|---|
| 1 | 5a | CH$_3$CN | 54% | 10:1 | 12% |
| 2 | 5b | CH$_3$CN | 68% | 14:1 | 8% |
| 3 | 5c | CH$_3$CN | 46% | 8:1 | 32% |
| 4 | 5d | CH$_3$CN | 59% | 16:1 | 26% |
| 5 | 5e | CH$_3$CN | 78% | 15:1 | 11% |
| 6 | 5f | CH$_3$CN | trace | n.d. | n.d. |
| 7 | 5g | CH$_3$CN | 79% | 12:1 | 74% |
| 8 | 5h | CH$_3$CN | 75% | 7:1 | 72% |
| 9 | 5g | n-BuCN | 81% | 14:1 | 69% |
| 10 | 5g | CH$_2$Cl$_2$ | 63% | 10:1 | 67% |
| 11 | 5g | THF | trace | n.d. | n.d. |
| 12 | 5g | MTBE | trace | n.d. | n.d. |
| 13[e] | 5g | CH$_3$CN | 82% | 19:1 | 89% |
| 14[f] | 5g | n-BuCN | 86% | 19:1 | 87% |
| 15[f] | 5g | CH$_3$CN/n-BuCN (5:1) | 85% | 20:1 | 92% |

Ad adamantyl, dr diastereomeric ratio, ee enantiomeric excess, n.d. not determined, n-BuCN n-Butyronitrile, MTBE methyl tert-butyl ether.
[a]Reaction conditions: 1a (0.2 mmol), 2a (0.1 mmol), 3a (0.1 mmol) and CPA 5 (5 mol%) in the solvent specified (4.0 mL) under irradiation using 440 nm Kessil LEDs at 0 °C for 48 h under argon.
[b]Isolated yields.
[c]Determined by $^1$H NMR analysis of the crude mixture.
[d]Determined by HPLC analysis on a chiral-stationary-phase.
[e]Reaction at -42 °C.
[f]Reaction at -50 °C.

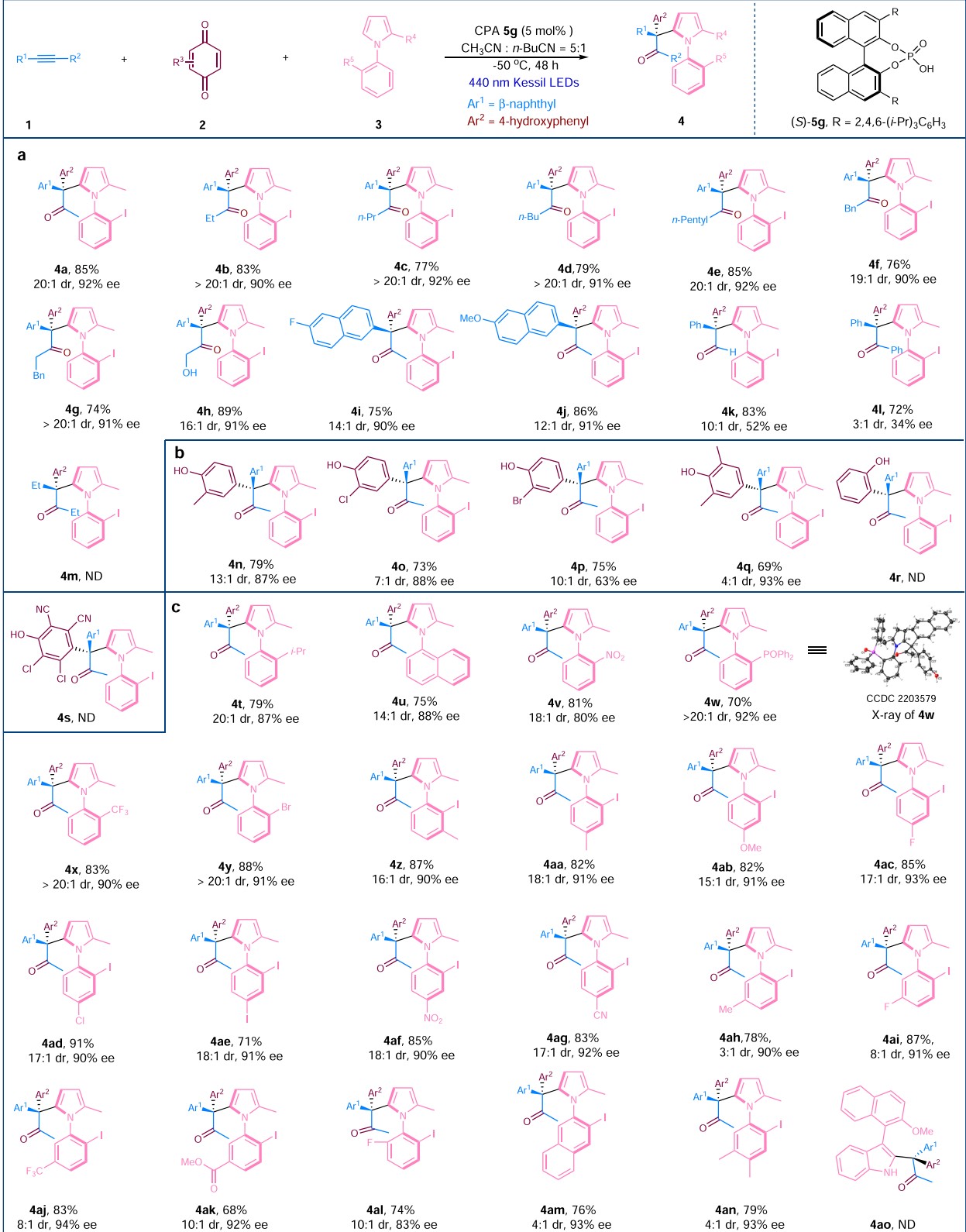

**Fig. 2 | Reaction scope.** Reaction conditions: **1a** (0.2 mmol), **2a** (0.1 mmol), **3a** (0.1 mmol) and CPA **5** (5 mol%) in CH₃CN/n-BuCN (v/v 5:1, 4.0 mL) under irradiation using 440 nm Kessil LEDs at −50 °C for 48 h under argon; isolated yields reported.

**a** The scope of the alkyne substrates. **b** The scope of the benzoquinone substrates. **c** The scope of the N-arylpyrrole substrates. ee enantiomeric excess, ND not detected, n-BuCN n-Butyronitrile.

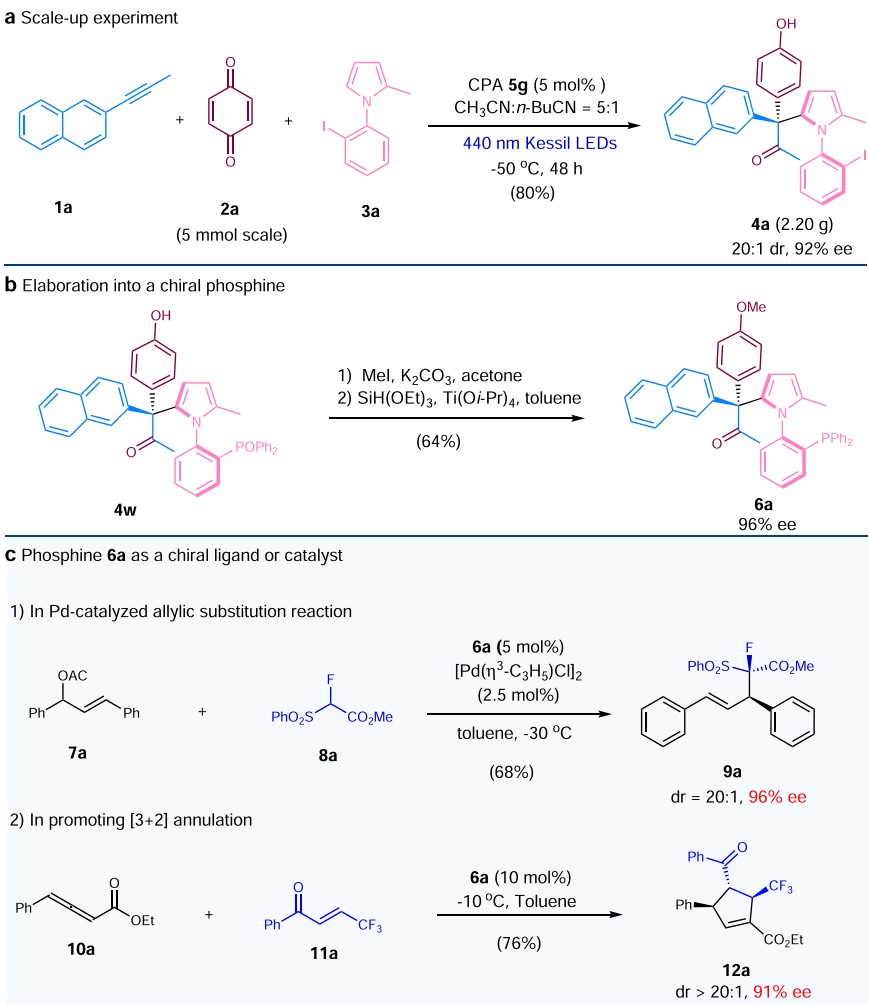

**Fig. 3 | Scale-up experiment and the applications of chiral *N*-arylpyrroles. a** Scale-up experiment. **b** Elaboration into a chiral phosphine. **c** Phosphine **6a** as a chiral ligand or catalyst.

employment of dialkyl alkynes, likely due to the low activity of *p*-QM intermediate (**4m**). The benzoquinone substrates could also be varied (Fig. 2b), and consistent good results were attainable (**4n–4q**). Nonetheless, when 1,2-benzoquinone or 2,3-dichloro-5,6-dicyano-*p*-benzoquinone (DDQ) was used, no desired products were observed (**4r** & **4s**).

The generality of the reaction with regard to the *N*-arylpyrrole substrates was next studied (Fig. 2c). Various *N*-arylpyrroles with a mono-substituted phenyl ring possessing electronically and sterically diverse functional groups were evaluated, and consistent high yields, excellent diastereoselectivities, and very good enantioselectivities were attainable (**4t, 4v–4y**). The reaction also worked for an *N*-naphthylpyrrole substrate (**4u**). Subsequently, the suitability of *N*-arylpyrroles bearing a disubstituted phenyl ring was examined. Regardless of the substitution patterns, i.e. *ortho-, meta-*, or *para-*, and electronic nature of the substituents, e.g. halogens, methyl/methoxyl, nitro, cyano or ester, the desired products were obtained in high yields, with excellent enantioselectivities and diastereoselectivities (**4z–4ak**). It is noteworthy that when the *N*-arylpyrrole containing 2,6-disubstituted phenyl moiety was employed, the challenging axially chiral product bearing four substituents along the C−N axial bond, including a quaternary stereogenic center, was prepared in good yield, with good enantioselectivity and excellent diastereoselectivity (**4al**). Finally, *N*-naphthylpyrrole and *N*-arylpyrroles bearing a trisubstituted phenyl ring were suitable for the reaction, and the yields and stereoselectivities were well-maintained (**4am** and **4an**). We also

examined 2-aryl indole as a potential nucleophile, and the desired product was not detected (**4ao**). The absolute configurations of the products were assigned on the basis of the X-ray crystallographic analysis of **4w** (see the Supplementary Information and Supplementary Tables S1–S6).

## Synthetic application

To showcase the practicability of our method, a scale-up experiment was performed, axially chiral **4a** was prepared in 80% yield with 92% ee (Fig. 3a). We felt at the outset that the *N*-arylpyrroles being constructed herein may be used as a chiral ligand or a catalyst in asymmetric catalysis, we thus proceeded to synthesize a chiral phosphine (**6a**, 96% ee) from one of axially chiral products (**4w**) (Fig. 3b). Notably, **6a** has a C−N axial chirality, as well as a central quaternary stereogenic center at the pyrrole 2-position. To our delight, **6a** was found to be a good ligand in palladium-catalyzed allylic substitution reaction[78,79], furnishing product **9a** in excellent diastereo- and enantioselectivities. Furthermore, **6a** also turned out to be an excellent chiral phosphine catalyst, promoting the [3 + 2] annulation[80] between allenoate **10a** and alkene **11a** in a highly stereoselective manner (Fig. 3c).

## Discussion
### Mechanistic studies
Preliminary mechanistic studies were performed (Fig. 4). The UV−vis spectra of alkyne **1a**, benzoquinone **2a** and *N*-arylpyrrole **3a** were

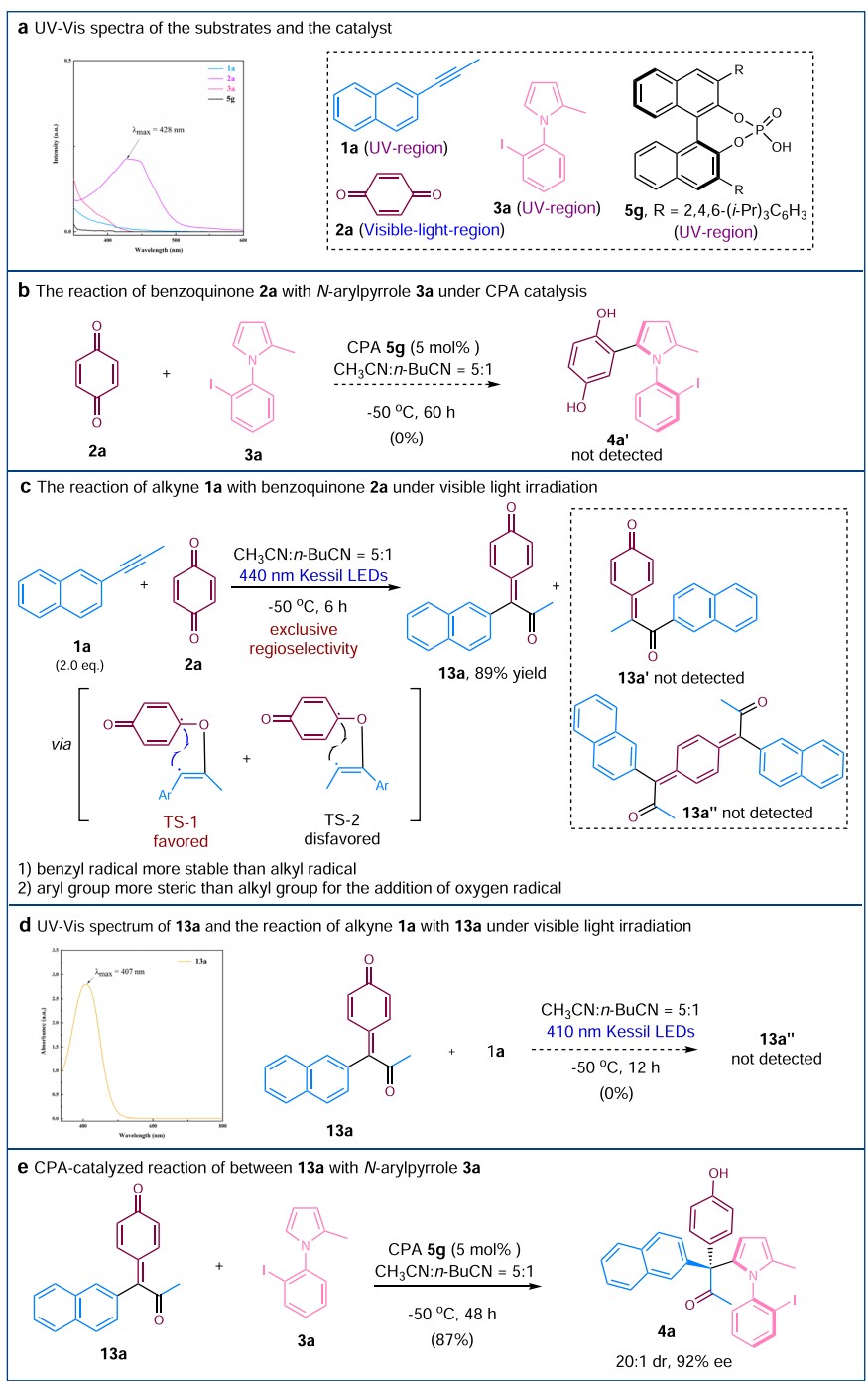

**Fig. 4 | Mechanistic studies. a** UV–Vis spectra of the substrates and the catalyst.
**b** The reaction of benzoquinone **2a** with *N*-arylpyrrole **3a** under CPA catalysis. **c** The
reaction of alkyne **1a** with benzoquinone **2a** under visible light irradiation. **d** UV–Vis

spectrum of **13a** and the reaction of alkyne **1a** with **13a** under visible light irradia-
tion. **e** CPA-catalyzed reaction of between **13a** with *N*-arylpyrrole **3a**.

acquired, and only **2a** showed strong absorption band at the visible
light region with the maximum absorption peak at around 428 nm
(Fig. 4a). The reaction between benzoquinone **2a** and *N*-arylpyrrole **3a**
under CPA catalysis did not yield the corresponding product (**4a′**),
indicating that there is no such background reaction (Fig. 4b). When
two molar equivalences of alkyne **1a** and benzoquinone **2a** were irra-
diated using 440 nm Kessil LEDs, *p*-QM **13a** was formed in 89% yield,
the other regioisomer **13a′** and double addition product **13a″** were not
observed – these results are consistent with the regioselectivity
observed in our reaction. The excellent regioselectivity observed in
our reaction is likely attributed to two factors: the higher stability of

the benzyl radical compared to the alkyl radical and the steric differ-
ence between an aryl group and an alkyl group (TS-1 vs. TS-2) (Fig. 4c).
The UV−vis spectrum of **13a** was acquired, revealing a strong absorp-
tion band at the visible light region with the maximum absorption peak
at around 407 nm. When alkyne **1a** and *p*-QM **13a** were irradiated using
410 nm Kessil LEDs, the double addition product **13a″** was not
observed (Fig. 4d). If *p*-QM **13a** was reacted with *N*-arylpyrrole **3a** in the
presence of CPA **5 g**, the same axially chiral product **4a** was obtained in
high yield with excellent dr and ee values (Fig. 4e), suggesting that
*p*-QM may likely be the reaction intermediate during our one-pot,
three-component catalytic process.

In summary, we have developed a highly atroposelective synthesis of *N*-arylpyrroles through a one-pot, three-component oxo-diarylation reaction enabled by light-induced phosphoric acid catalysis. By directly employing unactivated alkynes as one of the substrates, a good range of arylpyrroles were prepared in good yields, with high distereo- and enantioselectivities. Notably, the products contain both C−N axial chirality and a nearby central quaternary stereogenic center, which were simultaneously created in a highly stereoselective manner. Moreover, facile structural elaboration of the *N*-arylpyrrole product led to the formation of a chiral ligand/an organic catalyst which have been shown to be very useful in asymmetric catalysis. By making use of readily available feedstocks i.e. unactivated alkynes and developing an efficient oxo-diarylation process, we are disclosing a new strategy for the construction of axially chiral *N*-arylpyrroles, which represent structural motifs that may be used as ligands/catalysts in asymmetric catalysis. We believe the method reported herein has a general implication for practical synthesis of novel axially chiral molecular architectures with potential applications in asymmetric catalysis and synthesis.

## Methods

### General procedure for asymmetric oxo-diarylation reaction

To a dried and argon-filled 10 mL screw-cap vial equipped with a magnetic stir bar were added alkyne **1** (0.2 mmol), benzoquinone **2a** (0.1 mmol, 10.8 mg), *N*-arylpyrrole (0.1 mmol), CPA **5g** (5 mmol%) and $CH_3CN$/*n*-butyronitrile (v/v, 5:1, 4.0 mL). The mixture was then irradiated by 440 nm Kessil LEDs at −50 °C. The reaction mixture was concentrated under reduced pressure after 48 h and the residue was purified by column chromatography on silica gel to furnish the product.

## Data availability

The authors declare that the data supporting the findings of this study are available within the article and its Supplementary Information file. For experimental details and compound characterization data, see Supplementary Methods. For $^1H$ NMR, $^{13}C$ NMR and $^{31}P$ NMR spectra, see Supplementary Figs. 1–84. The X-ray crystallographic coordinates for structures reported in this study have been deposited at the Cambridge Crystallographic Data Centre (CCDC) under deposition number 2203579 (**4w**). These data can be obtained free of charge from The Cambridge Crystallographic Data Centre via www.ccdc.cam.ac.uk/structures.

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

## Acknowledgements

Y.L. thanks the Singapore National Research Foundation, Prime Minister's Office for the NRF Investigatorship Award (A-0004067-00-02), and the Ministry of Education (MOE) of Singapore (A-0008481-00-00) for generous financial support.

## Author contributions

L.D. designed and carried out the experiments. X.Z., J.G., X.D. and Q.H. participated in the synthesis of substrates. L.D. and Y.L. conceived the project and wrote the manuscript. Y.L. supervised the project.

## Competing interests

The authors declare not competing interests.
