## [Peer review file · Nature Communications]

REVIEWER COMMENTS

Reviewer #1 (Remarks to the Author):

The report by Prof. Lu and co-workers detailed the diastereo- and atroposelective synthesis of N-arylpyrroles through a three-component oxo-diarylation reaction enabled by light-induced CPA catalysis. This transformation simultaneously creates a C–N axial chirality and a central quaternary stereogenic center with high diastereoselectivities. The reaction also demonstrated good functional group tolerance for some sensitive substituents, such as POPh₂, NO₂, COOMe, CN and others. Importantly, the utility study proved the constructed skeleton to be useful in asymmetric catalysis. Compared with their previous work on central chirality (Sci. Adv. 2022, DOI: 10.1126/sciadv.add2574), this reaction is potentially more useful.

There are some other issues need to be considered:

1. In the substrate scope of alkynes, only aryl alkyl alkynes were investigated, which demonstrated good regioselectivity and stereoselectivity. How about other types of alkynes, such as dialkyl, diaryl and terminal alkynes?
2. As for the scope of benzoquinones, have 1,2-benzoquinones been tested? Is there any background reaction of benzoquinones with N-arylpyrroles observed? This method will be greatly improved if more types of heterocycles (e.g. indole) could be used as nucleophiles.
3. In Page 4, line 71 (Results and discussion part), N-arylindole 3a should be corrected to N-arylpyrrole 3a.
4. A recent very related work for the construction of axially chiral arylpyrroles from alkynes should be cited (Angew. Chem. Int. Ed. 2023, e202303670).

To sum up, the work by Lu and co-workers detailed the synthesis of N-arylpyrroles through a three-component strategy. The reaction showed wide scope, good functional group tolerance and good utility. I think it deserves to be published on Nature Communications after the above minor revisions.

Reviewer #2 (Remarks to the Author):

Dai and Lu reported a light-induced efficient protocol for the synthesis of N-arylpyrroles with high diastereo- and atroposelectivity. Due to the worthy of N-arylpyrrole motifs, this work is suitable to publish in Nat. Commun. after minor revision.

1. Why the mixture of MeCN and BuCN could improve the enantioselectivity?
 2. I would like to see more examples of 2, only two examples are presented in Fig 2. What about DDQ?
 3. Would you explain why only 13a was obtained? what's the reason for the selectivity?
 4. I am curious about the UV-Vis of 13a. Is this also in visible-light region?
 5. what happen if two equivalents of alkyne reacted with 2a? Could two [2+2] type products was generated? And could this used for the present reaction to get double addition product?
 6. Some errors in SI, for example: the ^1H NMR of 4h, the number of proton are too many, please check them carefully!!
- 4i,4p: ^{13}C NMR, there are coupling of F to C, please check the NMR carefully.

Point-to-Point Response to Reviewers' Comments (Manuscript ID: NCOMMS-23-17409)

Please take note that all the descriptive, positive comments of the reviewers are omitted, and only the reviewers' comments expressing their concerns/suggestions are listed below, which are followed by our responses. All the changes made in the revised manuscript as highlighted in yellow.

Revisions made in reply to Reviewer 1' comments:

- Comment #1: In the substrate scope of alkynes, only aryl alkyl alkynes were investigated, which demonstrated good regioselectivity and stereoselectivity. How about other types of alkynes, such as dialkyl, diaryl and terminal alkynes?
- Our response: We have investigated the reactivities of dialkyl, diaryl and terminal alkynes under standard reaction conditions. The employment of terminal and diaryl alkynes led to the formation of the desired products (**4k** & **4l**), while the reaction with dialkyl alkynes did not work (**4m**). We revised the manuscript accordingly to include the above results (see revised Fig. 2a and accompanying discussion in the text).
- Comment #2: As for the scope of benzoquinones, have 1,2-benzoquinones been tested? Is there any background reaction of benzoquinones with N-arylpyrroles observed? This method will be greatly improved if more types of heterocycles (e.g. indole) could be used as nucleophiles.
- Our response: 1) The reaction with 1,2-benzoquinone did not deliver the desired product (**4r**) (see revised Fig. 2b). 2) There is no background reaction occurring between benzoquinones and N-arylpyrroles (see revised Fig. 4b and accompanying text). 3) The reaction may be applicable to other heterocycles, e.g. indoles, however, careful tuning and screening the structures of substrates may be required. In a quick examination, we employed a 2-aryl indole as the nucleophile, but no desired product (**4ao**) was detected (see revised Fig. 2c and related text).
- Comment #3: In Page 4, line 71 (Results and discussion part), N-arylindole **3a** should be corrected to N-arylpyrrole **3a**.
- Our response: Error corrected.
- Comment #4: A recent very related work for the construction of axially chiral arylpyrroles from alkynes should be cited (Angew. Chem. Int. Ed. 2023, e202303670).

- Our response: The above related work has now been cited (ref. 67 in the revised manuscript).

Revisions made in reply to Reviewer 2' comments:

- Comment #1: Why the mixture of MeCN and BuCN could improve the enantioselectivity?
- Our response: When the reaction was performed in acetonitrile at -42 °C, 89% ee and 19:1 dr were obtained (entry 13). Since the melting point of acetonitrile is at -45 °C, we then used a mixture of acetonitrile and *n*-butyronitrile (melting point -112 °C) to further enhance stereoselectivity at lower reaction temperature. After some experimentations, we established the optimal reaction conditions; when the reaction was performed in a mixed solvent system (acetonitrile/*n*-butyronitrile = 5:1) at -50 °C, the desired product was obtained in 85% yield, with 20:1 dr and 92% ee (entry 15) (see highlighted text above Table 1).
- Comment #2: I would like to see more examples of **2**, only two examples are presented in Fig 2. What about DDQ?
- Our response: We have used different benzoquinone structures and included more such examples (see revised Fig. 2b, **4n-4q**). However, the utilization of DDQ did not lead to the formation of the desired product (**4s** in revised Fig. 2b).
- Comment #3: Would you explain why only **13a** was obtained? what's the reason for the selectivity?
- Our response: The excellent regioselectivity observed in our reaction is likely attributed to two factors: the higher stability of the benzyl radical compared to the alkyl radical and the steric difference between an aryl group and an alkyl group (see revised Fig. 4c, TS-1 vs. TS-2).
- Comment #4: I am curious about the UV-Vis of **13a**. Is this also in visible-light region?
Our response: The UV-vis spectrum of **13a** was acquired, revealing a strong absorption band at the visible light region with the maximum absorption peak at around 407 nm (see revised Fig. 4d).
- Comment #5: What happened if two equivalents of alkyne reacted with **2a**? Could two [2+2] type products were generated? And could this used for the present reaction to get double addition product?

Our response: The reaction of two equivalents of alkyne **1a** with benzoquinone **2a** did not give double addition product **13a''** (see revised Fig. 4c). Furthermore, alkyne **1a** and *p*-QM **13a** were irradiated using 410 nm Kessil LEDs, no double addition product **13a''** was formed (see Fig. 4d in the revised manuscript).

- Comment #6: Some errors in SI, for example: the ¹H NMR of **4h**, the number of proton are too many, please check them carefully!! **4i, 4p:** ¹³C NMR, there are coupling of F to C, please check the NMR carefully.
- Our response: We have re-checked the ¹H NMR of **4h** and have made the corrections. The ¹³C NMR spectra of **4i** and **4p** (now re-labeled as **4x**) have been double-checked and C–F coupling constants have been added (see the revised SI).

REVIEWERS' COMMENTS

Reviewer #1 (Remarks to the Author):

I appreciate the additional experiments that were carried out by the authors so as to address my comments on the original submission. I am happy with the changes made and recommend it for publication in Nature Communications.

Reviewer #2 (Remarks to the Author):

I am happy to see all the problems have been well addressed. This paper could be published as the revised version.